# Non-Incremental Bottom-Up Knowledge Compilation of Neuro-Answer Set Programs

## Abstract

Neuro-Probabilistic Answer Set Programming offers an intuitive and expressive framework for representing knowledge involving relations, non-determinism, logical constraints, and uncertainty-aware perception. Such a high expressivity comes at a significant computational cost. To mitigate that, Knowledge Compilation (KC) approaches translate the logic program into a logic circuit for which inference and learning can be performed efficiently. Top-down KC approaches employ an intermediary step of translating the logic program into a CNF propositional formula, before the actual KC step. This has the drawback of requiring the use of auxiliary variables and a fixed variable ordering. Bottom-up KC approaches instead construct a circuit representation compositionally, by employing circuit operations that represent the subparts of the logic program, without the need of auxiliary variables and allowing dynamic variable ordering. However, intermediary circuits can grow quite large even when the end circuit is succinct. In this work, we develop a non-incremental bottom-up KC strategy that provably and empirically reduces the size of the intermediary representations compared to its incremental counterpart. We explore heuristics for v-tree initialization and dynamic variable reordering. Experimental results show that our method achieves state-of-the-art performance for a large class of programs.

## 1 Introduction

Answer Set Programming (ASP) is a powerful declarative programming language for representing and solving combinatorial problems and commonsense reasoning (Eiter et al., 2009). In short, an ASP program conveniently describes a problem as a set of facts, if-then rules, disjunctions and higher order constructs such as aggregates (sum, count, etc), arithmetic expressions and inequalities.

Probabilistic Answer Set Programming (PASP) extends ASP with the ability to represent probabilistic uncertain knowledge (Poole, 1993; Baral et al., 2004; Cozman & Mauá, 2020). By associating probabilities to the output of neural classifiers, Neural PASP programs provide an elegant formalism for developing neuro-symbolic AI systems that combine the learning and sub-symbolic representation capabilities of deep neural networks with the precise and justifiable reasoning abilities of symbolic systems (Manhaeve et al., 2018; Yang et al., 2020; Geh et al., 2024). Importantly, such systems can be then trained end-to-end using a distance learning approach.

The main computational approach for inference and parameter learning with PASP involves translating the program into a propositional logic formula using Clark completion (Clark, 1977). This formula is then compiled into a tractable logic circuit (Fierens et al., 2015; Li et al., 2023; Totis et al., 2023; Azzolini & Riguzzi, 2023), a process known as Knowledge Compilation (KC) (Darwiche & Marquis, 2002). Tractable here means that the circuit satisfies certain properties that ensure that desired queries can be answered in linear time in the size of the circuit.

KC can generally be performed in two ways. Top-down approaches take a CNF formula as input and build a circuit as the trace of a DPLL procedure that enumerates models (Darwiche et al., 2004; Muise et al., 2012; Fierens et al., 2015; Eiter et al., 2024). This requires introducing auxiliary variables to avoid a blown up of the CNF representation and breaking cycles, which creates difficulties for variable selection heuristics and ultimately produce unnecessarily large circuits (Vlasselaer et al., 2014). Even though there are approaches to reduce the number of auxiliary variables introduced (Fandinno & Hecher, 2023), the resulting circuits can still be significantly larger than necessary.

Bottom-up KC, instead, builds a circuit compositionally, by translating rules into circuits that are then combined via circuit operations (Wang et al., 2024). This dispenses the need of auxiliary variables, and allows for dynamic circuit minimization, which can lead to more succinct encodings. In incremental bottom-up KC, one obtains the result circuit by sequentially conjoining an incumbent representation with a circuit representation of a rule. As de Colnet (2023) noted, such an approach can produce large intermediary circuits even when the resulting circuit is small. They proposed instead to adopt a *non-incremental* approach that decomposes a CNF formula into variable-disjoint components, compiles components separately, then conjoin them. Such an approach is proven to bound the maximum size of the intermediary circuits. de Colnet (2023) however assumed that the input is a CNF formula that can be decomposable into disjoint components.

The main contribution of this work is the non trivial task of developing a non-incremental bottom-up KC approach for PASP programs without translation into CNF (which would require adding auxiliary variables). To accomplish that, we develop a heuristic to decompose a PASP program into a disjoint set of rules that are separately translated into circuit and jointly conjoined. We extended the results by de Colnet (2023) to show theoretical linear upper bounds of the size of the intermediary circuits. Our bottom-up strategy also allows us to take advantage of more efficient encoding of ASP-specific constructs such as cardinality constraints (Vlasselaer et al., 2014).

## 2 BACKGROUND

We start with an overview of key concepts in Neuro–Probabilistic Answer Set Programming and Knowledge Compilation relevant to this work.

### 2.1 ANSWER SET PROGRAMMING

For simplicity, we consider only ground programs, since the semantics of ASP programs are defined on their grounded versions (Eiter et al., 2009). Hence, a simple *atom* is an expression $p(c_1, \ldots, c_m)$ where $p$ is a predicate name and each $c_i$ is a constant. A *cardinality atom* is of the form $l\{a_1, \ldots, a_n\}u$, where $l \leq u$ are integers and $a_i$ is a simple atom. Intuitively, they represent that at least $l$ and at most $u$ of the atoms $a_1, \ldots, a_n$ must be simultaneously true (Syrjänen & Niemelä, 2001). *Choice atoms* are written as $\{a_1, \ldots, a_n\}$, where each $a_i$ is an atom; they express that each subset of those atoms should be considered as a candidate solution. An ASP program is a finite set of *disjunctive rules*, written as:

$$a_1; \ldots; a_k :- b_1, \ldots, b_m, \text{not } c_1, \ldots, \text{not } c_n., \tag{1}$$

where each $a_i$, each $b_i$ and $c_i$ is an atom. The atoms $a_i$ form the head of the rule, $b_i$ are the positive body, and $c_i$ are the negative body (the positive and negative parts form the body of the rule). If a rule has an empty head (i.e., $k = 0$), it is called a(n integrity) *constraint*, representing a condition (the body) that must not be violated. If a rule has an empty body ($m = n$) and a single simple head atom it is called a *fact*. A rule with a single simple head and only simple atoms ($k = 1$) is a *normal rule*.

An *interpretation* $I$ is a subset of the atoms of the program. An interpretation satisfies an atom, expression or rule as in a classic propositional logic sense, for instance, if some atom of the body is false or if both the body and the head are satisfied by $I$. A *model* is an interpretation satisfying all rules. A model $I$ is *minimal* if there is no other model $J$ such that $J \subset I$. The reduct of a program $P$ w.r.t. an interpretation $I$, denoted by $P^I$, is the program obtained by removing all rules whose body is *not* satisfied by $I$, then removing the negative bodies of the remaining rules. An interpretation $I$ is a *stable model* iff it is a minimal model of $P^I$.

The *dependency graph* of a program is a directed graph whose nodes are the atoms appearing in the program and there is an edge $b \rightarrow a$ for each rule where $a$ appears in the head and $b$ is in the body. If $b$ appears negated, then we say the edge is negative otherwise the edge is positive. A normal program is *stratified* if it contains no directed cycle that goes through a negative edge. A stratified program has exactly one stable model; programs that can be broken into a stratified part and a set of integrality constraints have either 0 or 1 stable model. A normal program is *tight* if it contains no directed cycle that contains only positive edges. Tight programs can be translated into semantically equivalent normal programs in polytime (Linke et al., 2004).

The *Clark completion* (Clark, 1977) obtains a propositional formula that represents the supported models of the program by :

$$\bigwedge_{a \in \mathcal{A}(P)} \left[ a \Leftrightarrow \bigvee_{r \in \mathcal{R}(P,a)} \bigwedge_{b \in \text{body}(r)} b \wedge \bigwedge_{a' \in \text{head}(r) \backslash a} \neg a' \right],$$ (2)

where $\mathcal{A}(P)$ is the set of propositional atoms that appear in $P$, $\mathcal{R}(P,a)$ is the set of rules in $P$ that have $a$ as the head, $\text{body}(r)$ is the set of literals in the body of rule $r$, and $\text{head}(r)$ is the set of atoms in the head of rule $r$. Every stable model is a (propositional) model of the Clark completion but the converse is not necessarily true. It is true for example when the program is tight (Ben-Eliyahu & Dechter, 1994). In general, a stable model is a model of the Clark completion which also satisfies additional constraints known as *loop formulas* (Lee & Lifschitz, 2003). That property is used by many competitive ASP solvers to compute answer sets by a reduction to propositional satisfiability (Giunchiglia et al., 2006). As SAT solvers typically take CNF encodings as input, such an approach either resorts to incremental encodings or require the addition of a significant amount of auxiliary variables (e.g., worst-case quadratic in the number of atoms) to enable succinct CNF encodings. Also, loop formulas might incur in an exponential blow up in size (Lifschitz & Razborov, 2006).

## 2.2 NEURO-PROBABILISTIC ANSWER SET PROGRAMMING

One can extend an ASP program to cope with uncertainty by equipping it with *annotated disjunctions*, written as $\pi_1 :: a_1; \ldots; \pi_k :: a_k$, where $\pi_i$ are probabilities that sum to 1, and $a_i$ are simple atoms. Those probabilities can result from the outcome of a neural probabilistic classifier, thus connecting symbolic and subsymbolic representations. For our purposes here, however, we consider such probabilities as fixed parameters (the extension to end-to-end neural network learning is straightforward (Yang et al., 2020)). For simplicity, we assume that annotated disjunctions are disjoint, meaning that no atom appears in more than one annotated disjunction. We also assume that atoms in an annotated disjunction do not appear as heads of (non-probabilistic) rules. Under such assumptions, annotated disjunctions can be interpreted as representing categorical random variables, as follows. We also write $\pi :: a$ to denote a *probabilistic fact*, that is, an annotated disjunction $\pi :: a; 1 - \pi :: a'$, where $a'$ is some auxiliary atom not appearing in the program.

Let $\mathcal{D}(P)$ denote the set of annotated disjunctions in a PASP program $P$. A *total choice* $\theta$ is a mapping from each $\pi_1 :: a_1; \ldots, \pi_k :: a_k$ in $\mathcal{D}(P)$ to an atom $a_i$; let $\text{Pr}(a_i) = \pi_i$. Each total choice induces a (non-probabilistic) ASP $P_\theta$ formed by the rules of the program and facts $a$ for each $a \in \theta$. By assuming independence of choices, such a program is generated with probability (Taisuke, 1995):

$$\mathbb{P}(\theta) = \prod_{r \in \mathcal{D}(P)} \text{Pr}(\theta(r)).$$ (3)

Each generated ASP $P_\theta$ is associated with a set of answer sets $\Gamma(\theta)$. Because of the assumptions of disjointedness of annotated disjunctions, $\Gamma(\theta) \cap \Gamma(\theta')$ for $\theta \neq \theta'$. We follow most approaches and assume that $\Gamma(\theta)$ is non-empty for any $\theta$ (a condition called *consistency*). See (Totis et al., 2023) and (Mauá et al., 2024) for a discussions on lifting such a restriction.

The classification of programs according to their dependency graph extends to PASP programs, by simply including atoms in annotated disjunctions as nodes in the graph. Thus, a (consistent) stratified PASP program admits a unique extension of the probability of total choices $\text{Pr}(\theta)$ to the probability of answer sets of the induced programs $\bigcup_\theta \Gamma(\theta)$. Non-stratified PASP programs however admit more than one such extension. Two commonly adopted approaches are the: *credal semantics* Cozman & Mauá (2020) and the *maximum-entropy (maxent) semantics* (Baral et al., 2004; Totis et al., 2023). The credal semantics considers all possible extensions from $\text{Pr}(\theta)$ to the distribution of answer sets $\bigcup_\theta \Gamma(\theta)$. Inferences are then usually focused on the respective upper and lower probability bounds:

$$\underline{\mathbb{P}}(a) = \sum_{\theta : a \in \cap \Gamma(\theta)} \mathbb{P}(\theta), \qquad \overline{\mathbb{P}}(a) = \sum_{\theta : a \in \cup \Gamma(\theta)} \mathbb{P}(\theta),$$ (4)

where $a$ is an atom and the notation $\cap S$ ($\cup S$) denotes the conjunction (disjunction) of elements in $S$. The maxent semantics instead averages the probabilities:

$$\mathbb{P}(a) = \sum_{\theta} \sum_{I \in \Gamma(\theta) : a \in I} \frac{\mathbb{P}(\theta)}{|\Gamma(\theta)|}.$$ (5)

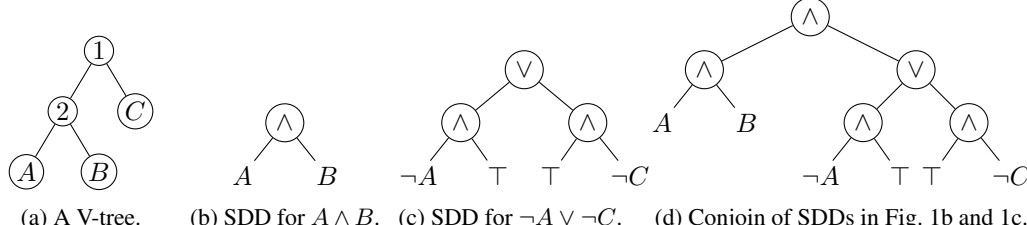

(a) A V-tree. (b) SDD for $A \wedge B$. (c) SDD for $\neg A \vee \neg C$. (d) Conjoin of SDDs in Fig. 1b and 1c.

Figure 1: Bottom-up compilation of $(A \wedge B) \wedge (\neg A \vee \neg C)$ into a str-DNNF that respects $V$-tree 1a.

### 2.3 KNOWLEDGE COMPILATION

Knowledge Compilation (KC) translates a propositional theory into a target representation for which certain inferences are performed efficiently. We focus on a specific class of target representations called *structured decomposable* Negation Normal Form (str-DNNF). Formally, a Negated Normal Form (NNF) is a rooted directed acyclic graph whose inner nodes denote either conjunctions (AND) or disjunctions (OR) and whose leaves are propositional literals. A Decomposable NNF (DNNF) satisfies *decomposability* of AND-nodes: the variable sets of any two input circuits are disjoint. Deterministic DNNF (d-DNNF) additionally satisfies *determinism* of OR-nodes, meaning that the logical formulae represented by any two input subcircuits are contradictory (i.e., have no common model). Finally, structured DNNFs (str-DNNF) satisfy *structural decomposability* (str-DNNF) of OR-nodes, which required the concept of a $V$-tree. Given set of variables $A$, a $V$-tree is a full, rooted binary tree whose leaves are in one-to-one correspondence with the variables in $A$ (Figure 1a). We say a decomposable circuit is str-DNNF if it respects a $V$-tree $T$. This means that for every conjunction $\alpha \wedge \beta$, there exists a node $t \in T$ such that the scope of $\alpha$ ($\beta$) is contained within the left (right) sub-tree of $t$. Sentential Decision Diagrams (SDDs) are a special case of str-DNNFs that (unlike general str-DNNFs) allow for efficient Boolean operations such as *Conjoin* (AND), *Disjoin* (OR), *NEG* (Negation) or *ITE* (If-Then-Else). This is at the core of bottom-up KC strategies, as exemplified in Figure 1.

Notably, d-DNNFs allow for efficient Weighted Model Counting (WMC):

$$\text{WMC}(\mathcal{C}) = \sum_{\omega \models \mathcal{C}} \prod_{\ell \in \omega} \text{weight}(\ell), \tag{6}$$

where $\mathcal{C}$ is a propositional formula (possibly represented as a circuit), the sum ranges over the models $\omega$ of $\mathcal{C}$, the product ranges over the literals $\ell$ in $\omega$, and weight($\ell$) is a nonnegative function.

When weight functions are associated to neural network outputs, WMC defines a loss function for learning under propositional logical constraints. Similarly, generalizations of WMC such as Algebraic Model Counting (AMC) (Kimmig et al., 2017) and Second Order Algebraic Model Counting (2AMC) Kiesel et al. (2022) capture probabilistic inference and gradient-based learning of Neural PASP programs under credal and maxent semantics (Wang et al., 2024).

## 3 RELATED WORK

NeurASP (Yang et al., 2020) extends ASP with neural predicates under the maxent semantics to provide a PASP-based neuro-symbolic framework. Inference and learning is performed by enumerating probabilistic choices and stable models (calling an external ASP solver), which limits their use to programs with few uncertain facts . dPASP Geh et al. (2024) extends NeurASP with the credal semantics and other generalizations (e.g., interval probabilities), but also uses an enumerative scheme to perform inference and learning.

In contrast, DeepProbLog (Manhaeve et al., 2018) and Scallop (Li et al., 2023) limit input to stratified PASP programs with neural predicates. Both the frameworks compile the program into SDDs using an incremental bottom-up compilation strategy, while differing in the approach to produce the formula (either Clark completion or forward chaining) (Fierens et al., 2015; Vlasselaer et al., 2014; 2016).

Totis et al. (2023) adopted a similar KC approach for non-stratified PASP programs under maxent semantics. However, their method uses top-down compilation with unconstrained sd-DNNFs as target language (Muise et al., 2012), followed by an enumerative re-normalization step. Consequently, this final enumerative step causes the approach to scale poorly with the number of answer sets.

Another top-down KC framework is described by Azzolini & Riguzzi (2023), who adapted the methods for Second Order Answer Set Model Counting by Eiter et al. (2024); Kiesel et al. (2022) to perform probabilistic inference under credal semantics. Their approach uses an X-first sd-DNNF as the target language. The role of different circuit properties, such as X-firstness, for various probabilistic semantics is further explored by Wang et al. (2024). They show that while X-firstness suffices for decision-dNNFs (Darwiche et al., 2004) and is implied by X-constrained SDDs (Oztok et al., 2016), it is neither necessary nor sufficient for maxent semantics in general.

To our knowledge, this is the first attempt to perform non-incremental bottom-up compilation of general PASP programs to generate succinct representations. Although SDDs have been used for this task by Eiter et al. (2024), their approach relies exclusively on top-down KC, which does not exploit the advantages of bottom-up compilation for generating more compact circuits without auxiliary variables.

## 4 BOTTOM-UP COMPILATION

We now present novel algorithms and techniques for non-incremental bottom-up PASP knowledge compilation. Our approach extends the bottom-up compilation method by Vlasselaer et al. (2014) to handle the more expressive stable model semantics with disjunctive rules, and integrity and cardinality constraints (Eiter et al., 2009); credal and maxent semantics of PASP can thus be implemented by imposing additional constraints during the compilation process (Wang et al., 2024). We assume that the input for our compilation process is a (ground) PASP program, where annotated disjunctions have been turned into choice rules. This is a common intermediary step among other compilers (Eiter et al., 2024; Azzolini & Riguzzi, 2023). The output is a circuit representation whose input is the atoms of the original program; this circuit can be then used to efficiently perform algebraic model counting inferences (including parameter learning) (Kimmig et al., 2017; Eiter et al., 2024).

### 4.1 COMPILATION

Adapting bottom-up compilation in stratified PASP (Vlasselaer et al., 2014) to non-stratified PASP semantics requires overcoming some challenges, such as disjunctions in the head and constraints. Additionally, the credal and maxent semantics require the compiled circuit to satisfy X-determinism (Wang et al., 2024). This is because the credal and maxent semantics implement a two-level AMC, unlike stratified PASP. This involves an inner semiring that counts over answer sets and an outer semiring that performs weighted model counting over the probabilities (Kiesel et al., 2022).

The original bottom-up algorithm of (Vlasselaer et al., 2014) does not consider disjunctive rules, since it uses the completion of an atom $a$ as $a \Leftrightarrow \bigvee_{r \in \mathcal{R}(P,a)} \bigwedge_{b \in \text{body}(r)} b$. Hence, to cope with such disjunctive rules, the main modification that must be made to the bottom-up algorithm is to use Eq. 2 as the completion. In particular, the Clark completion of Eq. 2 can also be applied to integrity constraints, as these can be interpreted as $\bot \iff \bigvee_{r \in \mathcal{R}(P,\bot)} \bigwedge_{b \in body(r)} b$, where $\mathcal{R}(P, \bot)$ represents the set of integrity constraints. Thus, constraints can be viewed as a "sub-case" of the algorithm.

Choice atoms are an essential part of PASP and they require special attention when performing bottom-up compilation. When an atom $a$ is neither a fact nor appears as head of any rules in the program, we find that its equation in the Clark completion is equivalent to $a \iff \bot$ and, thus, $a$ must be compiled to represent *false*. This is not the case for both annotated disjunctions (including probabilistic facts) and choice atoms. For these atoms, we should just not compile their respective Clark completion, since both can be either *true* or *false*, as both are candidates for a solution.

Another key challenge in PASP compilation is the representation of cardinality constraints. While there are well-studied methods in the literature capable of encoding such constraints more succinctly, such as Sequential Counters (Marques-Silva & Lynce, 2007) and Totalizers (Bailleux & Boufkhad, 2003), they introduce auxiliary variables, which our approach aims to mitigate. Thus, we propose a method to compile an upper cardinality constraint by calling $\text{Upper}(A, 0, u)$. This is an adaptation

of (Abío et al., 2012) that leverages bottom-up compilation to avoid introducing auxiliary variables:

$$\text{Upper}(A, c, t) = \begin{cases} \text{false,} & \text{if } c > t; \\ \text{true,} & \text{if } |A| \leq t - c; \quad (7) \\ \text{ITE}(\text{Upper}(A \setminus \{a\}, c + 1, t)), \text{Upper}(A \setminus \{a\}, c, t)), & \text{otherwise.} \end{cases}$$

where $A$ represents the set of atoms inside the cardinality constraint, $c$ is a counter of the current number of *true* atoms at the time of the function call, and $u$ is the upper value of the constraint. Even though Eq. 7 only encodes an upper cardinality constraint, it is fairly straightforward to generalize it for a lower bound or an "exactly $k$" constraint. The "exactly $k$" constraints are of special importance, since annotated disjunctions can be seen as "exactly one" cardinality constraints. Usually, annotated disjunctions are encoded using a method proposed by (Shterionov et al., 2015), which essentially encodes a Sequential Counter (Marques-Silva & Lynce, 2007) by introducing auxiliary variables.

Finally, if one were to compile a non-tight program, there are two approaches to cope with the positive cycles: cycle-breaking (Eiter et al., 2021); or compilation of loop-formulas (Lee & Lifschitz, 2003). The state of the art for top-down KC applies cycle-breaking algorithms in order to circumvent positive cycles in the program, due to the possible blow-up of the number of loop formulas. However, our experimental results show that, in classes of programs, directly compiling loop formulas might lead to more succinct circuits, since this approach does not introduce auxiliary variables.

Although the proposed method thus far results in a circuit that represents the answer sets of the underlying ASP program, it misrepresents the probabilistic semantics we consider do to lack of constraints (Wang et al., 2024). However, to correctly represent such semantics, we can constrain the circuit to be $X$-deterministic by restricting its V-tree to have its probabilistic variables on the left and the logical ones on the right. It is important important to note that this algorithm does not require the target language to be an SDD. The algorithm can be adapted to use any target language that supports both efficient bottom-up compilation, (weighted) model counting, and $X$-determinism. For example, by imposing $X$-determinism to str-DASCs (Onaka et al., 2025), one can obtain a compact representation for PASP programs that does not require the application of determinism to perform model counting (Onaka et al., 2025; Wang et al., 2024) for PASP semantics (Eq. 4 and 5).

### 4.2 V-Tree Optimization

The size of an SDD strongly depends on its V-tree. One of the key challenges in SDD compilation is finding a good V-tree that minimizes the size of the compiled circuit, without introducing a large computational overhead. Although research on good heuristics for circuit ordering initialization is a well-explored topic in the domain of CNFs (Darwiche, 2011), CNFs do not possess as well-structured relationships between their variables as is the case with Probabilistic Logic Programs (PLPs). Therefore, one of the proposals of this work is an heuristic for obtaining good variable orderings for V-tree generation, using the structure of the program to guide the search space.

The initialization heuristic for V-trees proposed in this work is based on the program; and can be employed in other PLPs (Fierens et al., 2015; Li et al., 2023). First, we construct the dependency graph of the program, where each node represents an atom and each edge represents a dependency between two atoms (whether it is a positive or negative dependency). Then we compute the number of descendants for each node in the graph. Finally, we sort the atoms in descending order of the number of descendants. The only restriction that we apply to PASP's case is that the V-tree must be $X$-constrained, so the probabilistic variables have precedence over the (logical) variables.

### 4.3 Non-Incremental Compilation

A key challenge in Knowledge Compilation are the intermediary circuits that are generated during the compilation process. Although certain formulas can be represented in a compact form, with polynomial size, the compilation process itself can lead to an exponential blow-up in the size of the circuit when compiling the program (de Colnet, 2023). Thus, we present a theorem that shows that PASP non-incremental compilation can lead to more efficient circuits, specially in cases where others approach would lead to exponential blow-up when compiling intermediate circuits. The core idea behind this approach is illustrated in Figure 2, where the compilation of $\Delta \wedge \Sigma$ is performed by dividing the compilation task into different *clusters*, that are independently compiled and then conjoined; as opposed to the standard incremental compilation, which linearly conjoins.

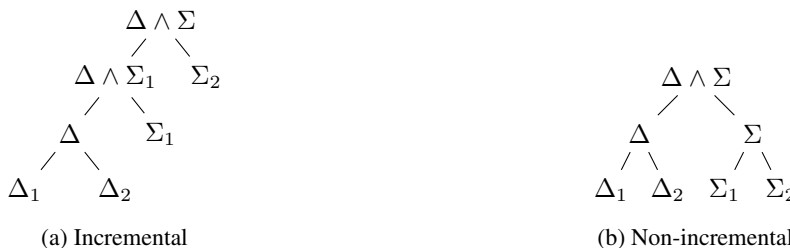

(a) Incremental                    (b) Non-incremental

Figure 2: Example of incremental and non-incremental approaches for compiling $\Delta \wedge \Sigma$.

In more detail, we create an undirected dependency graph of the Clark Completion of the program, were nodes are indexed by all the heads that are present in the program, and we have edges between nodes $u$ and $v$ iff they have an atom in common, either as a head or in the body. With this graph, we are able to detect disjoint subsets $c_1, \ldots, c_m$ of the Clark Completion rules by applying a connected components algorithm, such as Union-Find; and then compile each component $c_i$ into a circuit $\Delta_i$ by using bottom-up algorithm. Finally, we conjoin all $\Delta_i$ to obtain the representation of Eq. 2.

Although this approach works well for programs that are disjoint by nature, it can not be applied to programs that have only one connected component, which can be the case if one desires to express more complex PASP programs with inherent interdependencies. In these cases, we propose applying an algorithm to find a set of nodes that, when removed from the previous graph, will render the graph disconnected into disjoint components, allowing application of the non-incremental compilation. The only special consideration is that, after compiling the disjoint components into a circuit $\Delta$, one also needs to compile the logical constraints that were removed from the graph into another circuit $\Sigma$. The final circuit is then obtained by conjoining $\Delta \wedge \Sigma$.

This bottom-up algorithm is theoretically proven to reduce overall memory requirements by avoiding large intermediary circuits, which can render an incremental compilation process intractable (de Colnet, 2023). We formalize this argument in the following theorem:

**Theorem 1.** *Given a program $P$, we can determine $m$ disjoint subsets of the program in polynomial time, that can be non-incrementally compiled into an circuit of size at most $(m-1) + \sum_{i=1}^{m} |S_i|$, where $S_i$ is the size of the largest circuit obtained by compiling each subset using the bottom-up KC.*

*Proof (Sketch).* The poly-time complexity of determining the disjoint subsets is guaranteed by the use of a min-vertex-cut algorithm (Skiena, 1998). Since the variables in each $S_i$ are pairwise disjoint and we consider only str-DNNFs in when performing bottom-up compilation, Lemma 13 in (de Colnet, 2023) guarantees that the size of the final circuit is at most the sum of the maximum sizes of the intermediate circuits created during the compilation of each component $S_i$. □

## 5 EXPERIMENTS

**Infrastructure**  All experiments were conducted on a machine with a Ryzen 5 9600x CPU and 64GB of RAM, with a timeout of 30 minutes for each instance. For the top-down knowledge compilers, we used C2D (Darwiche et al., 2004) (with $dt\_method = 3$) and adaptations of D4 (Lagniez & Marquis, 2017) and SHARPSAT (Korhonen & Järvisalo, 2021) that enforce the necessary constraints for PASP inference (Eiter et al., 2024). The bottom-up compilation was implemented using the SDD library (Darwiche, 2011).

**Datasets**  For benchmarking the performance of our non-incremental algorithm, we propose using the benchmark proposed by (Azzolini & Riguzzi, 2024). We chose this benchmark over others, such as those from (Eiter et al., 2024; Kiesel & Eiter, 2023), because those works typically compare PASP KC techniques using CNF compilation benchmarks. Our approach, in contrast, focuses on directly compiling the program to the target representation, skipping the CNF translation step, which makes those benchmarks inapplicable. This benchmark corresponds to 4 classes of PASP programs: **Graph Coloring**; a non-stratified encoding of the **Smokers** dataset (Vlasselaer et al., 2014) (one of the most popular PLP programs in the literature); **IRL** and **IRN**, which corresponds, respectively,

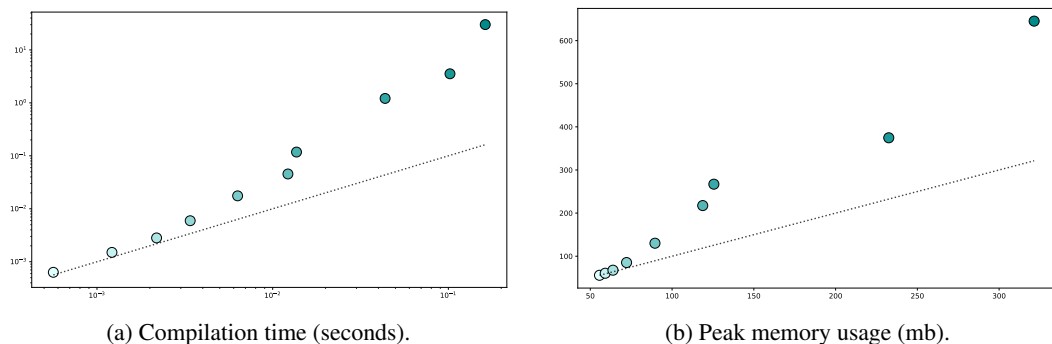

(a) Compilation time (seconds).
(b) Peak memory usage (mb).

Figure 3: Comparison between incremental (y-axis) and non-incremental (x-axis) compilation as we increase the number of nodes (darker colors) of the graph **Coloring** program. The dotted black line represents the baseline; points above it indicate that the incremental approach performed worse.

to classes of programs where we fix: the number of rules and increase the body size; fix the body size and increase the number of rules. To create random graphs for the graph coloring, we employ the approach of (Wang et al., 2020) alongside snowball sampling. For the non-stratified Smokers, we use a fully connected graph, as in the original bottom-up compilation article (Vlasselaer et al., 2014).

**Research Questions**    We analyze the results with respect to the following aspects.

**Q1: Does the non-incremental approach generate smaller intermediary circuits?** Figure 3 indicates that the non-incremental approach was able to use considerably less memory when compiling instances of the **Graph Coloring** program; and also compile larger instances (up to 15 nodes), whereas the incremental approach timed out on smaller instances (at 12 nodes).

**Q2: Does the proposed initialization heuristic generate more succinct representations?** Table 1 also supports the usefulness of the proposed heuristic, showing that it is distinct from other techniques in the literature. It was able to compile larger instances of the Coloring dataset in competitive time while producing more succinct intermediary representations.

**Q3: Is the bottom-up compilation of loop-formulas more succinct than cycle-breaking?** Now that we've confirmed that using both the non-incremental compilation and proposed heuristic can be beneficial, we tackle one of the most common assumptions on the PLP compilation: cycle-breaking is always the best choice (which was also employed in the original bottom-up compilation paper (Vlasselaer et al., 2014)). Table 2 shows that not applying cycle-breaking can result in significantly smaller compilation times, or even compiling a larger instance size. This loop formulas compilation is an advantage specific of bottom-up compilers, since they can circumvent costly translations of DNF to CNF when compiling loops.

| #Nodes | Paper | | MinDegree | | MinFill | |
|---|---|---|---|---|---|---|
| | mb | s | mb | s | mb | s |
| 13 | **374** | **2.42** | 10020 | 229 | 805 | 4.07 |
| 14 | **387** | **5.48** | - | - | 754 | 7.21 |
| 15 | **997** | 32.2 | - | - | 2939 | **31.99** |
| 16 | **9640** | **279** | - | - | - | - |
| 17 | **10411** | **588** | - | - | - | - |

Table 1: Comparison of memory (**mb**) and time (**s**econds) between the proposed heuristic, Min-Degree and MinFill for $V$-tree initialization, in the **Coloring** dataset for non-incremental KC.

| #People | Cycle | Loop | Cycle+Min | Loop+Min |
|---|---|---|---|---|
| 2 | **0.004** | **0.004** | 0.006 | 0.005 |
| 3 | 0.006 | **0.005** | 0.028 | 0.014 |
| 4 | 0.028 | **0.014** | 0.310 | 0.129 |
| 5 | 0.713 | **0.070** | 21.51 | 1.413 |
| 6 | 492.6 | **1.156** | - | 36.56 |
| 7 | - | **58.506** | - | - |

Table 2: Comparison of execution time (seconds) across instances of the **Smokers** program: with(out) dynamic minimization (Min), and for cycle-breaking or looping variants.

**Q4: The bottom-up compilation generates more succinct circuits than top-down compilers?** Finally, we analyze Figure 4, benchmarking the non-incremental bottom-up compilation against

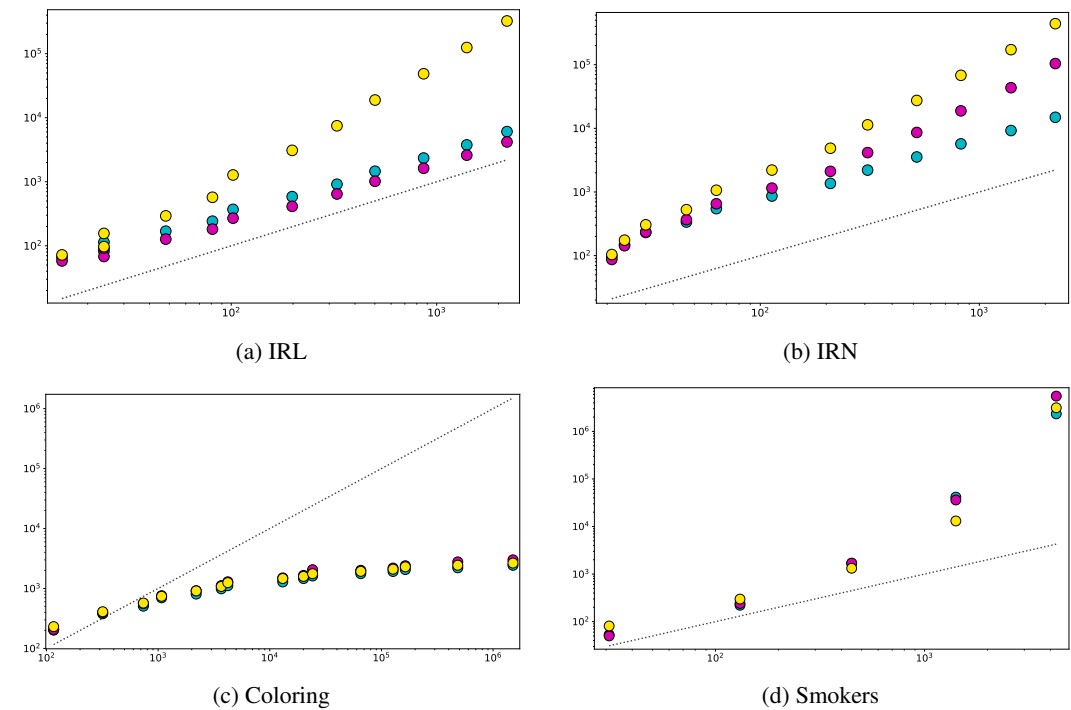

Figure 4: Succinctness comparison across the four datasets, with x-axis and y-axis representing the size of the circuit produced by the bottom-up and top-down compilers, respectively. Cyan, magenta and yellow represent, respectively: C2D, D4 and SHARPSAT-TD. The black dotted line acts as baseline: if a top-down compiler was placed above/below, it generated less/more succinct circuits.

the top-down approaches. Our initial expectation was that the bottom-up approach would excel in programs like **Smokers** and **IRN**, where atoms appear as heads of multiple rules, and perform worse on **Coloring** and **IRL**. The results confirm this expectation for **Smokers** and **IRN**, where our non-incremental approach was able to compile larger instances and generate considerably more succinct circuits than the top-down methods. Notably, our method also demonstrated superior performance on the **IRL** dataset, surpassing all top-down compilers. This unexpected success is attributed to our method's dynamic V-tree restructuring capability, which effectively optimized the compilation.

## 6 CONCLUSION

We've presented novel methods for non-incremental Probabilistic Answer Set Programming (PASP) Knowledge Compilation (KC), alongside theoretical results demonstrating their potential for efficient compilation. A key innovation in our approach lies in its non-incremental nature: we decompose the original program into disjoint subsets, compile each independently, and then conjoin their respective circuits to form the program's final representation. Furthermore, we've adapted bottom-up KC to effectively handle PASP-specific constraints, including cardinality constraints and probabilistic semantics. Overall, our methods demonstrate potential for significant improvements in the efficiency and scalability of PASP inference, because it avoids introducing auxiliary variables during compilation. By moving beyond standard top-down pipelines and exploring alternative circuit compilation, we enable the generation of considerably more intricate circuits. This enhanced compilation capability, in turn, allows neuro-symbolic AI systems to encode more complex real-world constraints. This deep integration of neural perception with robust PASP reasoning can be further leveraged by recent advances in the encoding of circuits on GPUs (Maene et al., 2024).

REPRODUCIBILITY STATEMENT

Our work is designed to be highly reproducible, with every effort made to ensure that our results can be independently verified. All theoretical claims are formally stated in the main paper, and any theorems or algorithms that build upon existing work are duly referenced to their original sources.

For computational experiments, we have included a comprehensive code package in the supplementary materials. This includes all source code for data preprocessing and running our experiments. A detailed README file provides step-by-step instructions for downloading and executing the top-down compilers used for comparison and running our own code. To further guarantee replicability, we have ensured the code is clear and well-commented, and we used fixed random seeds and a consistent dataset.

We have fully documented our experimental setup, including the computing infrastructure, hardware specifications, and the names of all relevant software libraries. All datasets used, including any novel ones, are fully described in the supplementary materials and will be made publicly available with a license that permits free usage for research purposes. While our primary analysis focuses on direct performance summaries, all necessary data and methods are provided for others to perform more detailed statistical analyses.

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

## A  APPENDIX

We report here details about the datasets used in the experiments. For each dataset, we provide the (non-grounded) program definition, a more in depth discussion about its motivation, and present results that may not be included in the main text, including tables and images where applicable.

The programs in Sections A.1, A.2, A.3 and A.4 were used in the main article, and their encodings were obtained from (Azzolini & Riguzzi, 2024). The programs in Sections A.5, A.6 and A.7, on the other hand, are from our own authorship.

### A.1  3-COLORING

This class of logic programs encodes 3-coloring graph problems, which consist in partitioning the nodes of a graph intro three sets such that no edge contains both endpoints in the same part.

PROGRAM DEFINITION

We use the same encoding as in (Azzolini & Riguzzi, 2024) for the 3-Coloring programs. To populate these programs, we create random graphs of various sizes (1 to 30 nodes) by applying snowball sampling to the Bitcoin OTC dataset (Kumar et al., 2016; 2018), a methodology similar to that of (Wang et al., 2020).

```
% Grounded Coloring Program
0.5::edge(X, Y). % Probability of edge between nodes X and Y
node(X). % Fact also derived from the dataset
% A node can have only one of three colors
red(X) :- node(X), not green(X), not blue(X).
green(X) :- node(X), not red(X), not blue(X).
blue(X) :- node(X), not red(X), not green(X).
% Symmetry between edges
e(X, Y) :- edge(X, Y).
e(X, Y) :- edge(Y, X).
% 3 graph coloring codification as
:- e(X, Y), red(X), red(Y).
:- e(X, Y), green(X), green(Y).
:- e(X, Y), blue(X), blue(Y).
```

RESULTS

Here, we present the results regarding Q1 of the main paper in larger plots, to facilitate visualization. Figures 5a and 5b show a comparison between incremental (y-axis) and non-incremental (x-axis) compilation, while Table 3 presents the same analysis via a table.

Regarding Q2 of the article, Figure 4 presents a more in depth comparison between the different heuristics used for the graph coloring problem, this time including the impact on circuit size. Note that, while the Min Fill heuristic was able to consistently generate smaller circuits, it was not able to compile instances that the proposed heuristic could.

Finally, regarding Q4, Figure 6 shows, in larger scale, a comparison between top-down and bottom-up compilation.

### A.2  PIN (NON-STRATIFIED SMOKERS)

The PIN dataset models the dynamics of a disease spread across contact network. Individuals can get infected either by contact with other *infected* individual of the network or by an external event (in-

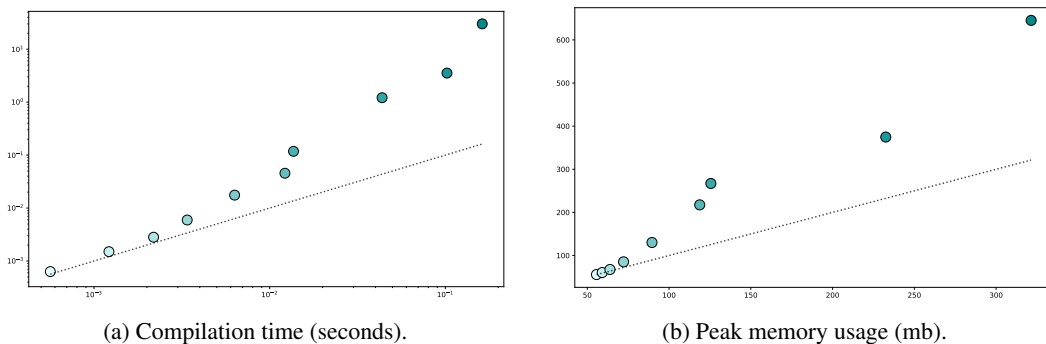

(a) Compilation time (seconds).          (b) Peak memory usage (mb).

Figure 5: Comparison between incremental (y-axis) and non-incremental (x-axis) compilation as we increase instance size (darker colors) of the graph **Coloring** program. The dotted black line represents the baseline for the non-incremental approach (being above it is worse).

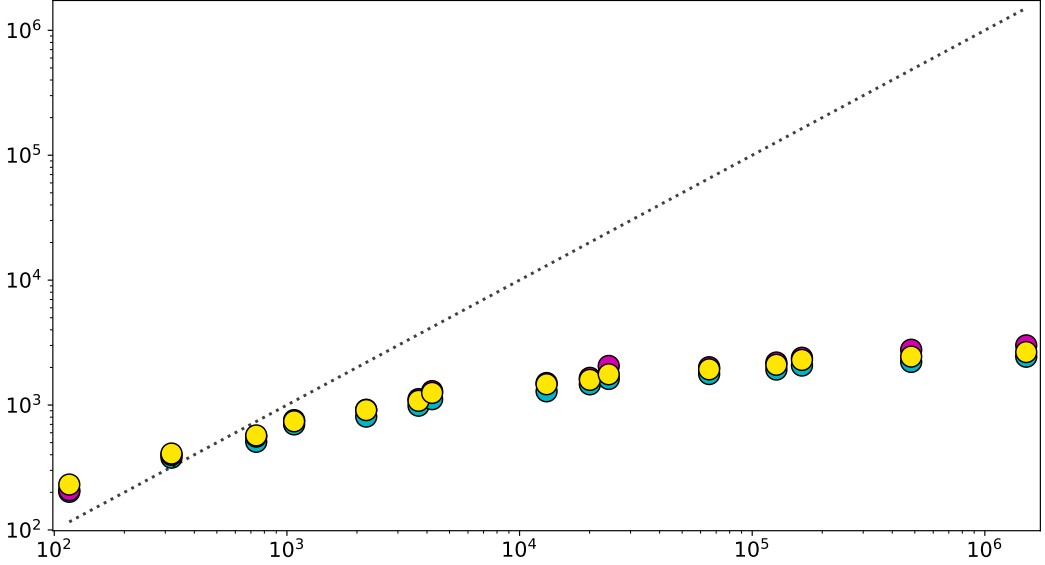

Figure 6: Performance comparison on the Coloring dataset, with x-axis representing the size of the circuit produced by the bottom-up compiler and the y-axis by the top-down approaches. Cyan, magenta and yellow represent, respectively: C2D, D4 and SHARPSAT-TD. The black dotted line acts as baseline: if a top-down compiler was placed above, it generated less succinct than the bottom-up; if below, otherwise.

Table 3: Memory and Time Results for Bottom-Up Compilation Approaches for the 3-Coloring dataset.

| #Nodes | Incremental | | Non-Incremental | | Non-Incremental+Heuristic | |
|---|---|---|---|---|---|---|
| | Time (s) | Memory (MB) | Time (s) | Memory (MB) | Time (s) | Memory (MB) |
| 2 | 0.00063 | 55.81 | **0.00056** | **55.63** | 0.00056 | 66.43 |
| 3 | 0.00153 | 60.53 | **0.00121** | **59.14** | 0.00126 | 59.17 |
| 4 | 0.00281 | 67.61 | 0.00218 | 63.85 | **0.0203** | **63.80** |
| 5 | 0.0062 | 85.32 | 0.00339 | **72.17** | **0.01741** | 130.16 |
| 6 | 0.01741 | 130.16 | 0.00631 | 89.54 | **0.00622** | **89.23** |
| 7 | 0.04606 | 217.46 | **0.01221** | 118.74 | 0.01292 | **118.21** |
| 8 | 0.11868 | 267.05 | 0.01366 | 125.56 | **0.01358** | **125.48** |
| 9 | 1.2354 | 374.65 | **0.04364** | 232.51 | 0.04499 | **232.05** |
| 10 | 3.54144 | 645.18 | **0.10215** | **321.42** | 0.10524 | 321.46 |
| 11 | 30.07836 | 2049.47 | **0.16216** | 344.08 | 0.17608 | **343.30** |
| 12 | - | - | **0.70257** | 273.18 | 0.7225 | **272.5312** |
| 13 | - | - | **2.36851** | 374.75 | 2.4229 | **374.47** |
| 14 | - | - | **5.45878** | **387.08** | 5.4767 | 387.26 |
| 15 | - | - | **32.06292** | 998.11 | 32.2010 | **997.81** |
| 16 | - | - | 272.25984 | 9640.56 | 278.54 | **9640.16** |
| 17 | - | - | **584.34** | 10411.66 | 588.93 | **10411.08** |

Table 4: Comparison of memory (**mb**), time (**seconds**), and circuit size ($\#nodes + \#edges$) between the proposed heuristic for $V$-tree initialization, Min Degree and Min Fill, in the 3-Coloring dataset for non-incremental compilation (without dynamic minimization). Instances with timeout are represented with "-".

| #Nodes | Proposed | | | MinDegree | | | MinFill | | |
|---|---|---|---|---|---|---|---|---|---|
| | mb | s | Size | mb | s | Size | mb | s | Size |
| 10 | 321 | 0.11 | 20051 | 367 | 0.62 | 20807 | **298** | **0.09** | **13157** |
| 11 | 343 | **0.18** | 24172 | 650 | 1.95 | **21941** | **234** | 0.24 | 26334 |
| 12 | **272** | **0.72** | 65303 | 1608 | 6.91 | **53071** | 385 | 1.17 | 64370 |
| 13 | **374** | **2.42** | 127032 | 10020 | 229 | 143930 | 805 | 4.07 | **129018** |
| 14 | **387** | **5.48** | 163629 | - | - | - | 754 | 7.21 | **153725** |
| 15 | **997** | 32.2 | 482389 | - | - | - | 2939 | **31.99** | **301560** |
| 16 | **9640** | **279** | 1506435 | - | - | - | - | - | - |
| 17 | **10411** | **588** | 1741436 | - | - | - | - | - | - |

Table 5: Circuit size comparison (in terms of number of nodes and edge in the circuit) for the 3-Coloring Problem.

| #Nodes | Non-Inc | | C2D | | D4 | | SharpSAT-TD | |
|---|---|---|---|---|---|---|---|---|
| | Node Size | Edge Size | Node Size | Edge Size | Node Size | Edge Size | Node Size | Edge Size |
| 2 | 37 | 79 | **73** | **129** | 99 | 109 | 101 | 130 |
| 3 | 98 | 221 | **132** | **247** | 187 | 212 | 173 | 236 |
| 4 | 227 | 511 | **173** | **335** | 262 | 297 | 241 | 330 |
| 5 | 328 | 746 | **237** | **463** | 353 | 404 | 311 | 430 |
| 6 | 660 | 1532 | **270** | **538** | 428 | 489 | 381 | 530 |
| 7 | 1054 | 2625 | **329** | **659** | 519 | 596 | 451 | 630 |
| 8 | 1255 | 2954 | **370** | **744** | 602 | 690 | 521 | 730 |
| 9 | 3511 | 9556 | **426** | **862** | 694 | 806 | 604 | 863 |
| 10 | 5575 | 14476 | **484** | **981** | 768 | 882 | 661 | 930 |
| 11 | 6860 | 17312 | **534** | **1084** | 960 | 1096 | 730 | 1033 |
| 12 | 18078 | 47225 | **584** | **1187** | 934 | 1074 | 801 | 1130 |
| 13 | 34881 | 92151 | **634** | **1290** | 1017 | 1172 | 871 | 1230 |
| 14 | 45917 | 117712 | **681** | **1390** | 1109 | 1282 | 948 | 1349 |
| 15 | 129208 | 353181 | **729** | **1486** | 1292 | 1480 | 1010 | 1433 |
| 16 | 400509 | 1105926 | **802** | **1635** | 1400 | 1603 | 1096 | 1559 |
| 17 | 493948 | 1301488 | **826** | **1686** | 1358 | 1586 | 1166 | 1665 |
| 18 | - | - | **879** | **1795** | 1539 | 1764 | 1220 | 1733 |
| 19 | - | - | **946** | **1935** | 1540 | 1775 | 1307 | 1856 |
| 20 | - | - | **982** | **2007** | 1596 | 1840 | 1361 | 1930 |

dicated by the probabilistic predicate *contaminated*). An infected individual might be *symptomatic* or not; non symptomatic individuals are called *vectors* of the disease.

This type of network displays the typical transitivity closure often used to evaluate logic program inferences (like the Smokers dataset). Relative to the Smokers program, this program contains also challenges relative to non-stratified negation and cyclic dependencies, as the vector and symptomatic contradictory nature increase considerably the complexity of the inference process, requiring $X$-determinism to ensure tractability.

PROGRAM DEFINITION

Unlike the 3-Graph Coloring dataset, we use a fully connected graph for the non-stratified Smokers program, following the methodology of the original bottom-up compilation article (Vlasselaer et al., 2014).

```
% Probabilistic Interaction Network
0.5::contaminated(1..N).
0.5::friend(1..N, 1..N).
infected(X) :- contaminated(X).
infected(X) :- friend(X, Y), infected(Y).
healthy(X) :- not infected(X).
symptomatic(X) :- infected(X), not vector(X).
vector(X) :- infected(X), not symptomatic(X).
```

RESULTS

Another interesting research question (call it **Q4**) that was not explored in the main paper is the following: Is the bottom-up compilation of loop-formulas more succinct than cycle-breaking? This can be answered by Table 6, where we can see that using cycle-breaking is not always the best choic. It is easy to see that directly compiling the positive cycles via loop formulas can result in significantly smaller compilation times, or even allow the compilation of larger instance sizes. This loop formulas compilation is an advantage specific of bottom-up compilers, since the bottom-up approach can more easily circumvent the costly translations of DNF to CNF when compiling loops.

Table 7: Cycle Breaking and Loop Formulas Comparison for PIN Dataset

| #People | Cycle Breaking Information | | | |
|---|---|---|---|---|
| | Loop Formulas | Aux Atoms | Aux Head Rules | Aux Body Rules |
| 1 | 0 | 0 | 0 | 0 |
| 2 | 1 | 2 | 2 | 2 |
| 3 | 5 | 9 | 15 | 12 |
| 4 | 20 | 28 | 64 | 48 |
| 5 | 84 | 75 | 215 | 160 |
| 6 | 409 | 186 | 636 | 480 |
| 7 | 2365 | 441 | 1743 | 1344 |
| 8 | 16064 | 1016 | 4544 | 3584 |
| 9 | 125664 | 2295 | 11439 | 9216 |
| 10 | 1112073 | 5110 | 28060 | 23040 |

Table 8: Circuit sizes for the PIN dataset across different configurations. Dashes indicate time-out.

| #People | C2d | | D4 | | Bottom-Up | | SharpSAT-TD | |
|---|---|---|---|---|---|---|---|---|
| | #Edges | s | #Edges | s | #Edges | s | #Edges | s |
| 1 | 28 | 0.002 | 26 | **0.0001** | **21** | 0.0043 | 41 | 1.0014 |
| 2 | 134 | 0.002 | **127** | **0.0002** | 159 | 0.0046 | 170 | 1.0014 |
| 3 | 1273 | 0.002 | 885 | **0.0006** | **707** | 0.006 | 887 | 1.0029 |
| 4 | 34601 | 0.015 | 18875 | **0.0137** | **5570** | 0.0274 | 10061 | 1.0153 |
| 5 | 1995714 | 1.336 | 2835725 | 2.6578 | **116184** | **0.7363** | 2777131 | 1.4128 |
| 6 | – | – | – | – | **5590** | **1.1560** | – | – |
| 7 | – | – | – | – | **29299** | **58.5064** | – | – |

| #People | Cycle | Loop | Cycle+Min | Loop+Min |
|---|---|---|---|---|
| 2 | **0.004** | **0.004** | 0.006 | 0.005 |
| 3 | 0.006 | **0.005** | 0.028 | 0.014 |
| 4 | 0.028 | **0.014** | 0.310 | 0.129 |
| 5 | 0.713 | **0.070** | 21.51 | 1.413 |
| 6 | 492.6 | **1.156** | - | 36.56 |
| 7 | - | **58.506** | - | - |

Table 6: Comparison of execution time (seconds) across instances of the **Smokers** program: with(out) dynamic minimization (Min), and for cycle-breaking or looping variants.

Also regarding Q4, Table 7 shows that, even though the number of loop formulas can greatly increase (specially in this fully connected example of the PIN dataset), the number of auxiliary atoms and rules introduced by cycle-breaking also increases significantly. This possibly explains why the top-down approaches are not able to compete with the bottom-up compiler in terms of circuit size when trying to compile larger instances (of size 6 or 7).

Similarly to the previous section, Figure 7 and Table 8 show, in larger scale and depth, that the top-down approaches are not able to compete with the bottom-up compiler in terms of circuit size and inference time.

### A.3    IRL

The IRL dataset represents a sequence of probabilistic facts and logical rules. It is designed to test the scalability of encoding techniques with respect to the size of the body of the rules, with a fixed number of rules in the program. It is a fairly simple dataset, that acts a baseline and should, in theory,

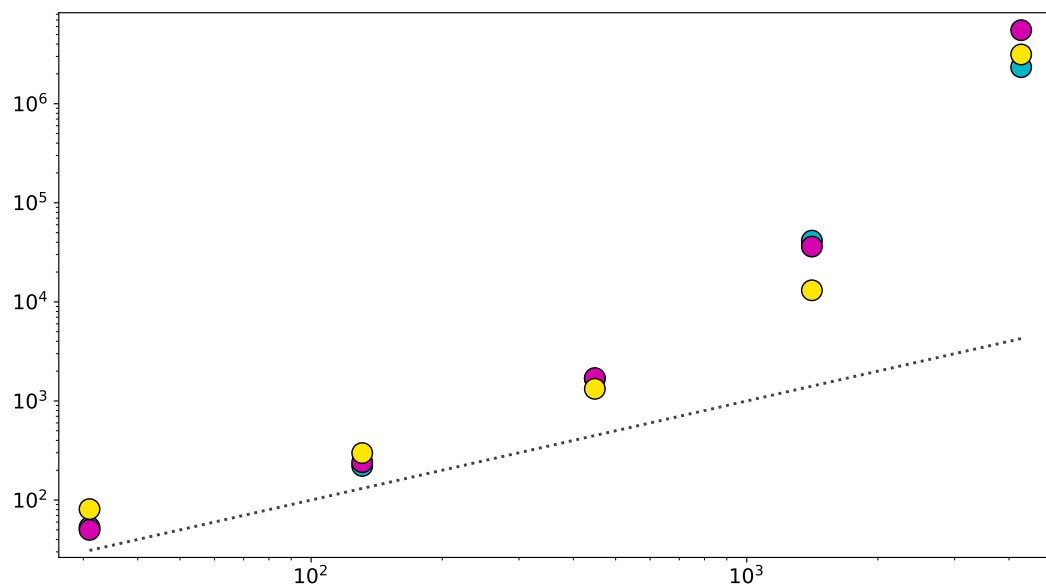

Figure 7: Performance comparison on the PIN dataset, with x-axis representing the size of the circuit produced by the bottom-up compiler and the y-axis by the top-down approaches. Cyan, magenta and yellow represent, respectively: C2D, D4 and SHARPSAT-TD. The black dotted line acts as baseline: if a top-down compiler was placed above, it generated less succinct than the bottom-up; if below, otherwise.

behave specially well for top-down compilers, since there is no need to introduce auxiliary variables in the Clark completion.

PROGRAM DEFINITION

Given the simplicity of the IRL dataset (Azzolini & Riguzzi, 2024), we varied the parameter $N$ from 1 to 500.

```
% IRL Problem
0.5::a(1..N).
qr :- a(X0), a(X2), a(X4), ..., a(Xeven).
qr :- a(X1), a(X3), ..., a(Xodd), not nqr.
nqr :- a(X1), a(X2), ..., a(Xn), not qr.
```

RESULTS

Again, Figure 8 shows in larger scale that the top-down compilers C2D and D4 have an slight advantage over the non-incremental bottom-up approach, while the SHARPSAT-TD was consistently worse than the other approaches.

## A.4 IRN

Similarly to the IRL dataset, the IRN dataset is designed to test the scalability of encoding techniques with respect to the number of rules in the program, with bodies of fixed (almost) unitary length. the body of the rules. It is more complicated than the IRL dataset, since an atom can happen due to multiple rules being satisfied, and there are many rules creating negative cycles between both $qr$ and $nqr$. This is an example where the bottom-up approach should be more efficient than the top-down approach, due to the increasing number of auxiliary variables required to encode the Clark completion.

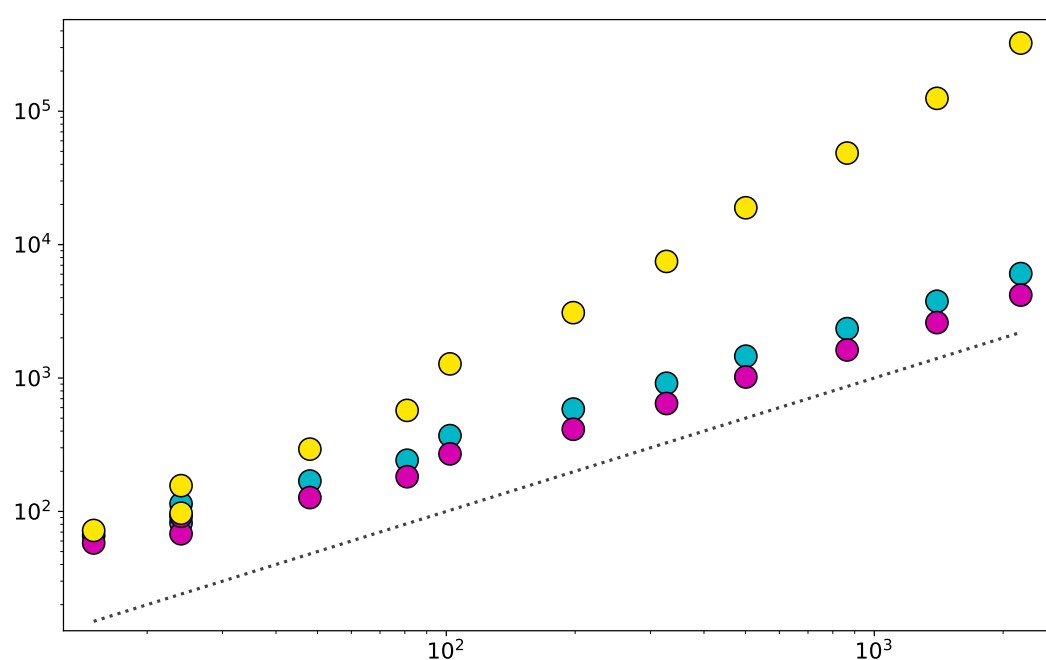

Figure 8: Performance comparison on the IRL dataset, with x-axis representing the size of the circuit produced by the bottom-up compiler and the y-axis by the top-down approaches. Cyan, magenta and yellow represent, respectively: C2D, D4 and SHARPSAT-TD. The black dotted line acts as baseline: if a top-down compiler was placed above, it generated less succinct than the bottom-up; if below, otherwise.

PROGRAM DEFINITION

For the IRN dataset (Azzolini & Riguzzi, 2024), we employed a similar strategy, varying the parameter $N$ from 1 to 500.

```
% IRN Problem
0.5::a(1..N).
qr :- a(Xeven).
qr :- a(Xodd), not nqr.
nqr :- a(Xodd), not qr.
```

RESULTS

Figure 9 shows the performance comparison on the IRN dataset, where we have attenuated results, with the top-down approaches being considerably worse than the bottom-up approach.

A.5   N-QUEENS

The N-Queens dataset models a probabilistic version of the classical N-Queens problem, where queens must not attack each other. It demonstrates the effectiveness of encoding techniques in handling spatial constraints. Due to the high number of possible conflicts, it is expected that representing this problem via a top-down approach will lead to quick blowups in circuit sizes.

PROGRAM DEFINITION

For the $N$-Queens dataset, we vary the number of queens from 1 to 13.

```
rows(1..N).
% N-Queens Problem
```

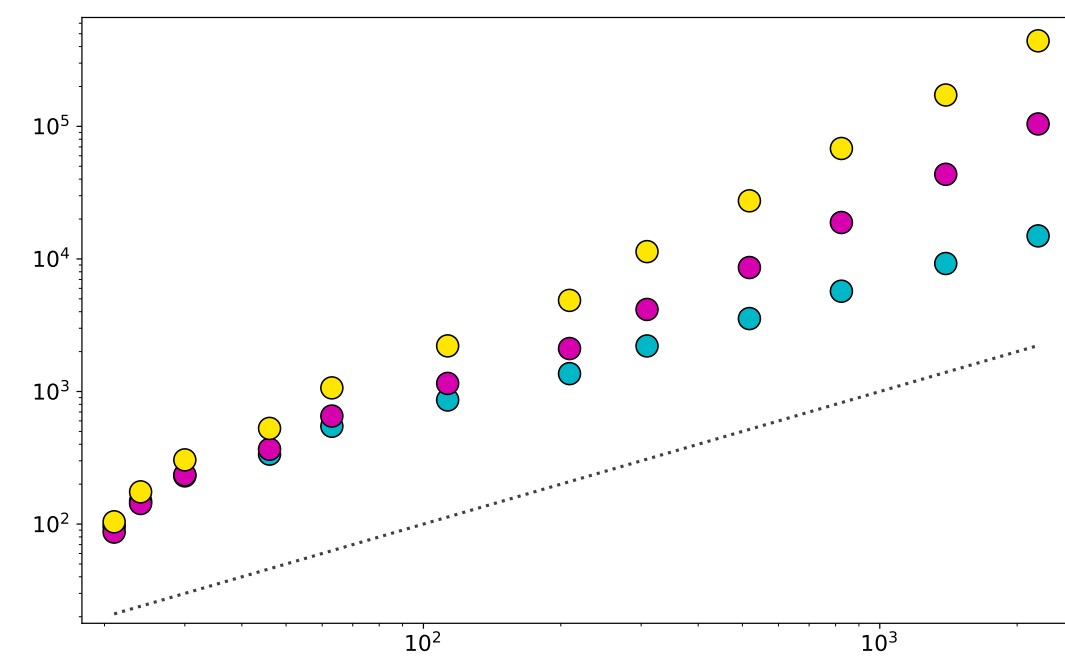

Figure 9: Performance comparison on the IRN dataset, with x-axis representing the size of the circuit produced by the bottom-up compiler and the y-axis by the top-down approaches. Cyan, magenta and yellow represent, respectively: C2D, D4 and SHARPSAT-TD. The black dotted line acts as baseline: if a top-down compiler was placed above, it generated less succinct than the bottom-up; if below, otherwise.

```
% For each row, there is a queen with a random
% distribution over the columns.
1/N::queen(R, 1), ..., 1/N::queen(R, N).
% We encode both satisfiable and unsatisfiable instances
% so there should always be a model
conflict :- queen(R1, C1), queen(R2, C2), abs(R1 - R2) == abs(C1 - C2).
conflict :- queen(R, C1), queen(R, C2), C1 != C2.
conflict :- queen(R1, C), queen(R2, C), R1 != R2.
```

We note that another common encoding of this problem uses cardinality constraints of the type "exactly one queen per row/column/diagonal". This encoding favors even more the proposed bottom-up KC approach by dispensing with the need of auxiliary variables.

RESULTS

The probabilistic version of the N-Queens problem encoding both satisfiability and unsatisfiability constraints is shown to be more efficiently encoded using a bottom-up approach, as one can see in Table 9. The bottom-up configuration was able to compute instances of double the size of the top-down ones in less than a second, and further tests shown that the bottom-up approach was able to compute instances of triple the size (12 queens) in less than 12 minutes.

A.6 FOOD

The Food dataset represents a preference selection problem, where the majority needs to decide on item to be selected (the typy of food they will have). This dataset is used to evaluate the impact of different encodings of cardinality constraints including constraints other than "exactly-one-of". The number of food can be fixed in order to vary the number of people and, thus, increasing only the cardinality constraints for the majority vote and annotated disjunctions.

Table 9: Circuit sizes for the N-Queens dataset across different configurations. Dashes "–" indicate time-out.

| #Queens | C2D | | D4 | | Bottom-Up | | SharpsAT-TD | |
|---|---|---|---|---|---|---|---|---|
| | #Edges | s | #Edges | s | #Edges | s | #Edges | s |
| 2 | 110 | 0.002 | 83 | **0.000102** | 18 | 0.004346 | 121 | 1.001351 |
| 3 | 2731 | 0.004 | 2946 | **0.000821** | 91 | 0.004507 | 2890 | 1.002867 |
| 4 | 134796 | 0.13 | 789178 | 0.147791 | 326 | **0.005276** | 152153 | 1.025344 |
| 5 | – | – | – | – | **414** | **0.009839** | – | – |
| 6 | – | – | – | – | **1110** | **0.020791** | – | – |
| 7 | – | – | – | – | **3417** | **0.074062** | – | – |
| 8 | – | – | – | – | **10098** | **0.225670** | – | – |

PROGRAM DEFINITION

When creating instances of the Food dataset, we kept the number of food items, $M$, constant as 4. This effectively fixed the number of voting options and allowed us to study the effects of varying the number of voters, $N$, from 1 to 25.

```
% Food Preference Problem
% Define the domains of people and food
person(1..N). food(1..M).
% Annotated disjunctions encode preferences
1/M::prefers(P,1); ...; 1/M::prefers(P,M) :- person(P).
% Exactly one food type must be chosen
1 { chosen(F) : food(F) } 1.
% Someone agrees if their prefered food is chosen
agrees(P) :- person(P), prefers(P,F), chosen(F).
% Constraint: More than half must agree
:- { P : agrees(P) } n//2.
```

RESULTS

The analysis of the impact of increasing the complexity of cardinality constraints can be seen in Table 10, where we've fixex the number of foods (parameter $M$) as 4. If one were to use the standard encoding of the program, unrolling cardinality constraints, without any optimizations (like Sequential Counters or Totalizers), both the C2D and D4 could only compile up to instances with 14 people; while the SHARPSAT-TD could only compile up to instances with 12 people. By using better encodings techniques, the top-down compilers were able to compile more instances, but the resulting circuit size was considerably larger than the one produced by the bottom-up compiler.

## A.7 HMM

The HMM dataset models a hidden Markov model with probabilistic facts and logical rules. It evaluates the ability of encoding techniques to handle sequential dependencies. It is very similar to the IRN dataset, with small bodies of rules, and an increasing number of rules. The main difference is that atoms may appear at most two times as heads of rules, which can be very advantageous for top-compilers, since their Clark completion auxiliary variables increase linearly with instance size.

PROGRAM DEFINITION

For the HMM dataset, we systematically varied the size of the underlying Hidden Markov Chain from 1 to 16.

```
% HMM Problem
0.5::a(1..N). 0.5::b(1..N).
x(1) :- a(1).
```

Table 10: Circuit size comparison for increasing problem size as measured by number of edges for the Food Dataset. For each compiler, we selected the best encoding technique: Bottom-Up used unrolling all constraints, as sequential counter and totalizer encodings performed worse; C2D used sequential counters; D4 and SharpSAT-TD used totalizer encodings. Dashes indicate time-out.

| #People | Bottom-Up | C2D | D4 | SharpSAT-TD |
|---|---|---|---|---|
| 16 | **6180** | 902711 | 188505744 | 172898394 |
| 17 | **8121** | 1165241 | – | 159943522 |
| 18 | **11591** | 1482106 | – | 40013987 |
| 19 | **11185** | 1867561 | – | 71239591 |
| 20 | **15249** | 2319095 | – | 232070177 |
| 21 | **15956** | 2864934 | – | 1624658814 |
| 22 | **16372** | 3495186 | – | 1451328317 |
| 23 | **22618** | 4244445 | – | – |
| 24 | **19731** | 5106888 | – | – |
| 25 | **25319** | 6118494 | – | – |

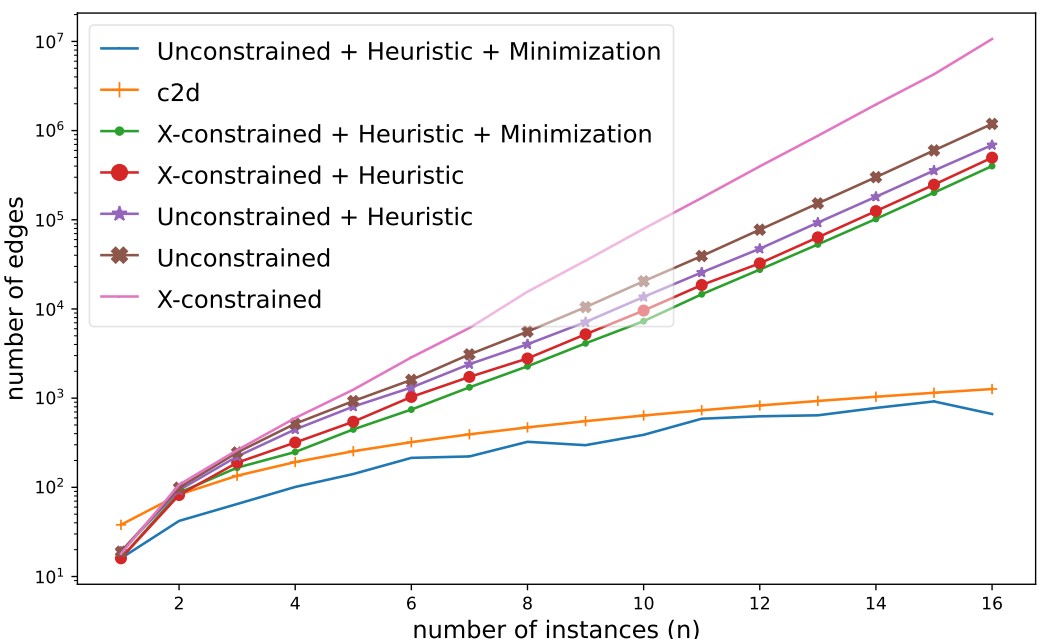

Figure 10: Performance comparison on the HMM dataset, with x-axis representing the instance size (size of the HMM chain) and the y-axis the number of edges in the circuits produced by the compilers.

```
x(X)  :- a(X), x(X-1).
y(X, 0)  :- x(X), not y(X, 1).
y(X, 1)  :- x(X), not y(X, 0).
```

## RESULTS

Finally, we present the results of our experiments on the HMM dataset. We compare the performance of the $c2d$ compiler with the bottom-up compiler, under a vast range of configurations, in order to show the impact of: imposing an X-constrained V-tree, using the proposed heuristic and how a top-down compiler can benefit from a program that has few auxiliary variables being introduced due to the low number of atoms appearing as heads of multiple rules.

