# OpenReview forum: "Non-Incremental Bottom-Up Knowledge Compilation of Neuro-Answer Set Programs"
_ICLR.cc/2026/Conference — Submitted to ICLR 2026_

### Official Review · Reviewer_LhmE · 2025-10-26

**Soundness:** 3
**Presentation:** 2
**Contribution:** 2
**Rating:** 2
**Confidence:** 4

**Summary:**

Knowledge compilation approaches in probabilistic answer set programming (PASP) can be categorised into top-down or bottom-up approaches.
Top-down typically require a CNF as input, which generally means additional auxiliary variables are introduced to first transform the PASP program into a CNF.
Bottom-up approaches do not have this requirement as they incrementally built up the compiled representation. During this process, intermediate representations may grow exponentially large. This work
1) develops a non-incremental bottom-up knowledge compilation strategy to reduce the size of the intermediate representations, and
2) explores a vtree initialization heuristic with dynamic variable ordering.

**Strengths:**

The work is relatively easy to read and well structured. Only, for the part about credal and maxent semantics I wonder if this is necessary for the story as it may just convolute it; and the related work deserves to be expanded.

The studied problem and contributions are very relevant for the knowledge compilation domain. Non-incremental bottom-up compilation is about not necessarily following the input circuit structure and instead optimising its compilation process to avoid intermediate blow-up of the representation size, which is an additional challenge for bottom-up compilation as opposed to top-down approaches. The vtree heuristic is important to obtain more succinct SDD representations in general. Both contributions are likely applicable beyond the probabilistic ASP setting.

The work is sound and the empirical results appear positive.

**Weaknesses:**

The paper title mentions Neuro ASP, but the contributions and experiments are not associated with neurosymbolic AI. While it is true that you could technically come up with a neurosymbolic version, this aspect is irrelevant to the contribution.

My main concern is novelty. The main contribution, to perform non-incremental compilation on rules, is not so different from non-incremental compilation on a CNF (de Colnet 2023). In both cases a set of variable disjoint components is identified and compiled separately. If such disjoint components are not present, some nodes are identified for removal such that disjoint components arise. This has parallels to variable ordering heuristics in top down compilers like D4, which favor branching on variables that would lead to independent components (Lagniez et al. 2017).

The related work discussion refers to other ASP related systems. However, the proposed vtree heuristic and the non-incremental decompositioning approach also relates to existing work, the connections of which are not discussed (e.g., minfill, mindegree, D4's branching heuristc, MINCE, ...). This is important to accurately assess novelty and position the work.

The importance and novelty of the cardinality constraint encoding, the choice atom, and disjunctive rules encoding are not clear. The latter two at least risk being trivial in the context of the existing work?

The work compares to top-down compilers, which require the introduction of auxiliary variables to obtain a CNF (cf. Tseitin procedure). However, it is worth noting that there is recent work that adapt a top-down compiler such that it does not require a CNF as input [Derkinderen et al. 2025]. They do perform a CNF translation internally, but then optimise the DPLL/CDCL search procedure to prune irrelevant artifacts that appear because of the auxiliary variables. Orthogonal but similarly, [Derkinderen 2024] proposed a post-compilation approach to again remove the auxiliary variables and irrelevant artifacts. So in terms of final representation size for top-down compilation, these works would improve the situation for top-down compilation as there are no more auxiliary variables and \iff structures afterwards.

[Derkinderen et al. 2025] Circuit-Aware d-DNNF Compilation. IJCAI 2025

[Derkinderen 2024] Pruning Boolean d-DNNF Circuits Through Tseitin-Awareness

**Questions:**

Q1) Sect 4.2 does not explain how to go from an ordered list of atoms to a vtree. Is this a left linear vtree that is then optimised using dynamic optimization? What were the used parameters for the dynamic optimisation process?

Q2) For top-down compilers, is the conversion to CNF part of the reported run time? And if so, how much time did this take usually?


**Suggestions**

The abstract mentions that top-down KC approaches require a fixed variable ordering. This is inaccurate as top-down compilers like sharpSAT-TD and D4 dynamically decide the variable branching order per branching decision. This statement probably refers specifically to top down SDD compiler.

Sect 4.2 states that "CNFs do no possess as well-structured relationships between their variables as is the case with Probabilistic Logic Programs (PLPs)". The meaning of this statement is not entirely clear to me. The primal graph of a CNF, a structural representation of the relationship between the variables, is commonly used within variable ordering heuristics like minfill?

The x/y tick label fontsize for Figure 3/4 are relatively small.

Typo:
* appendix "best choic."
* appendix p16 "Another interesting research question (call it Q4) that was not explored in the main paper is the following: Is the bottom-up compilation of loop formulas more succinct than cycle-breaking?" -- This research question was moved to the main paper.

---

### Official Review · Reviewer_Yy1K · 2025-10-27

**Soundness:** 2
**Presentation:** 2
**Contribution:** 2
**Rating:** 2
**Confidence:** 2

**Summary:**

This paper introduces a non-incremental bottom-up knowledge compilation (KC) strategy for probabilistic answer set programming (PASP), targeting neuro-symbolic reasoning systems. Traditional incremental bottom-up compilation suffers from large intermediate circuits even when final circuits are compact. The authors propose to partition PASP programs into variable-disjoint subcomponents, compile each separately, and conjoin the results, theoretically bounding intermediate circuit size. They also present a heuristic for V-tree initialization based on dependency graph structure, enabling dynamic variable ordering. Experiments on four PASP benchmarks (Coloring, Smokers, IRL, IRN) show improvements in memory and compilation time compared to incremental and top-down compilers (C2D, D4, SHARPSAT-TD).

**Strengths:**

1. The work addresses a genuine inefficiency in bottom-up PASP compilation (large intermediate circuits).

**Weaknesses:**

1. **Lack of neuro-symbolic integration.** Although framed as “Neuro-Answer Set Programs”, the paper’s experiments and formulations never demonstrate integration with neural components or end-to-end learning.
2. **Incremental vs. non-incremental framing.** The “non-incremental” decomposition heavily depends on detecting disjoint rule subsets or vertex cuts in the dependency graph. For complex or highly connected programs, the method degrades to incremental compilation. This limitation is only briefly acknowledged and lacks more detailed evaluation.
3. **Limited novelty relative to de Colnet (2023).** The core idea—compiling variable-disjoint components separately—seems to be directly inspired by de Colnet’s non-incremental strategy. The extension to PASP (without CNF) seems to be somewhat incremental.
4. **Insufficient Evaluation.**
    - Experiments are confined to small, synthetic datasets (≤17 nodes for Coloring) and classical Smokers/IRL examples. No large-scale or real neuro-symbolic applications are tested.
    - Performance comparisons omit recent bottom-up probabilistic frameworks such as Scallop (Li et al., 2023) and dPASP (Geh et al., 2024) beyond citation.
    - The empirical “state of the art” claim may be somewhat overstated: benchmarks are narrow and focused on circuit size, without inference accuracy etc.
5. **Regarding efficiency.** While the paper claims better efficiency over incremental KC, in Table 3 it shows very marginal improvement of the proposed non-incremental / non-incremental with heuristic, against the incremental baseline.
6. **Presentation issues.** The paper is somewhat hard to follow. Some sections (e.g., 4.1–4.3) read as an algorithmic sketch without pseudocode or formal definition of the decomposition algorithm, and it is not straightforward to understand which are existing techniques and which are the proposed novelty. Also, many of the result figures (Figure 3,4,6 etc.) have a somewhat unusual design and are a little hard to interpret. Also, the main theorem (L353) is stated but not proved in full detail (only a sketch at L357).

### **Minor**

1. I feel the background section may be too dense and verbose. I would recommend to summarize the background to involve less details and to be more straightforward, maybe within one page. The details can be moved to appendix.
2. There are some typos. e.g. L292 repeated word "important", L336 "were" should be "where".

**Questions:**

1. Could the authors clarify how does the 'neural' component come in to play in the proposed framework (as the title mentions "Neuro-ASP")? Does the proposed non-incremental compilation approach supports differentiable or neural integration in practice?
2. It seems the “non-incremental” approach relies on detecting variable-disjoint subprograms. If so, how does the method behave on highly connected PASP programs where such disjointness is rare?

---

### Official Review · Reviewer_rAAa · 2025-10-30

**Soundness:** 3
**Presentation:** 3
**Contribution:** 2
**Rating:** 6
**Confidence:** 3

**Summary:**

The paper describes an approach to perform efficient knowledge compilation for probabilistic answer set programs (PASPs). The traditional approaches require conversion to CNFs which introduces many auxiliary variables and increases complexity. Further, in existing approaches, the intermediate circuits generated during incremental compilation may be too large to be handled. The proposed approach presents a heuristic for compilation as well as compiling in an incremental manner to avoid exponential blow-up of circuit size. Experiments are performed on 3 benchmark problems and compared with existing state-of-the-art showing superior performance in terms of computational efficiency and resulting circuit size.

**Strengths:**

+ Seems to be a generalizable approach for compilation of PASPs with guarantees on compiled circuit size
+ Experiments seem show show a novel insight regarding cycle breaking, where they show that using this might result in larger circuit size
+ The results seem to show that the proposed approach performs better in terms of time/memory efficiency, intermediate and final circuit size when compared to existing methods

**Weaknesses:**

- The approach depends upon disjoint subsets in the program. In real-world scenarios do PASP programs typically have such subsets. In general, it would have been nice to show this encoding on a more realistic benchmark to show practical relevance. From the experiments it seems like the method bottoms out at nodes = 17 (table 1), is this scalable enough to be significant?
- The statement around why CNF based bottom-up methods/benchmarks comparison was inapplicable (line 376) is not so clear. If the goal is to show the proposed approach is an improvement over existing bottom up compilation methods (including CNF based ones), would that not make for a good comparison? Maybe some explanation is needed here.

**Questions:**

Why are existing bottom-up compilation methods excluded from the comparison?
What is the practical viability in terms of scaling up the approach?

---

### Meta-Review · Area_Chair_tyDy · 2025-12-23

**Summary:**

The reviewers raised several issues, such as novelty of the proposition, incorrect positioning of the work as neural-symbolic, or experimental validation issues.

**Reviewer Concerns:**

There is no rebuttal.

**Reviewer Scores:**

There is no rebuttal.

---

### Decision · Program_Chairs · 2026-01-26

Reject